# Socioeconomic inequalities in the risk factors of noncommunicable diseases (hypertension and diabetes) among Bangladeshi population: Evidence based on population level data analysis

**Md. Ashfikur Rahman** [ORCID] *

Development Studies Discipline, Social Science School, Khulna University, Khulna, Bangladesh

* ashfikur@ku.ac.bd

## Abstract

### Background

Noncommunicable diseases (NCDs) such as hypertension and diabetes are among the most fatal disease and prevalent among the adult population worldwide, including Bangladesh, and pose a public health threat. Understanding the socioeconomic inequalities linked to NCD risk factors can aid in the development of effective strategies to reduce the disease's recurrence. However, the literature on socioeconomic inequalities in hypertension and diabetes prevalence in Bangladesh is scant. Therefore, this study seeks to assess the inequality in hypertension and diabetes prevalence and to identify factors that may contribute to socioeconomic inequalities in Bangladesh.

### Methods

The current study incorporated data from a recent round of Bangladesh Demographic and Health Survey (BDHS 2017–18). The age-standardized prevalence rates of hypertension and diabetes were reported, and the log-binomial regression technique was used to identify the relevant confounders. Additionally, socioeconomic inequalities were quantified using a regression-based decomposition technique in which the concentration index (CIX) and Concentration curve were produced to determine the socioeconomic factors contributing to inequality.

### Results

Hypertension and diabetes were shown to have an age-standardized prevalence of (11.29% 95% CI: 11.13–11.69) and (36.98% 95% CI: 36.79–37.16), respectively. Both hypertension and diabetes were shown to be pointedly linked to the respondents' age, wealth status, being overweight or obese, and a variety of respondents' administrative divisions (p <0.001). In Bangladesh, household wealth status accounted for approximately 25.71% and 43.41% of total inequality in hypertension and diabetes, respectively. While

**Data Availability Statement:** This study used publicly available Demographic and Health Surveys Program datasets from Bangladesh, which can be

freely obtained from https://dhsprogram.com/. As a third party users the authors do not have permission to share the data.

**Funding:** The author(s) received no specific funding for this work.

**Competing interests:** The authors have declared that no competing interests exist.

BMI played a significant role in the emergence of inequality, the corresponding percentages for diabetes and hypertension are 4.95 and 83.38, respectively. In addition, urban areas contributed 4.56% inequality to increase diabetes among Bangladeshi inhabitants while administrative region contributed 4.76% of the inequality of hypertension.

## Conclusion

A large proportion of Bangladesh's adult population suffers from hypertension and diabetes. It is critical to recognize the value of equity-based initiatives in order to optimize the benefit-risk ratio and cost effectiveness of preventive health programmes. Integrating equity considerations into interventions is critical for policies and programmes to achieve their objectives. As a result, these findings can be taken into account when making existing and prospective policy decisions, as well as following its progression with economic development of Bangladesh.

## Background

The United Nations and other international organizations promote and seek integrated approaches to the Sustainable Development Goals (SDGs), which attempt to address current global health inequalities through multi-sectoral maneuvers [1]. Five SDGs include explicit aims for reducing health disparities on a national and global scale. These objectives include poverty eradication, universal health and well-being, equitable education, gender equality, and the decrease of inequalities both within and between countries. The reduction of health inequalities and poverty reduction is the prominent promise of SDGs [2]. Therefore, addressing the health inequalities is important and it is gaining policy recognitions globally including developing country like Bangladesh.

The biggest threat to public health in the twenty-first century has continued to be noncommunicable diseases (NCDs), which result in ill health, mortality, and disability as well as economic loss, loss of life, declining living standards, and poor social development in both high- and low-income countries (LMICs) [3–5]. NCDs are expected to kill 41 million people each year, accounting for 71% of all mortality globally, with 77% of those deaths occurring in LMICs including Bangladesh [3–6]. Various infectious diseases are still common in Bangladesh, and they have long been the most significant contributors to disease burden, while the burden of noncommunicable diseases (NCDs) is also rising [7, 8] which is an ultimate threat to fulfil the country's attainment of (SDG: 3.4). Which calls for 30% reduction in NCDs related mortality in the Global Action Plan for the Prevention and Control of NCDs through "prevention and treatment and promote mental health and well-being" [9, 10].

NCDs are becoming more prevalent as a result of socioeconomic inequalities not only in Bangladesh but also internationally [11]. NCDs share a number of behavioral risk factors, such as poor diet, excessive alcohol use, cigarette use, and sedentary lifestyles, all of which contribute to metabolic risk factors such as overweight and obesity, high blood pressure, high blood glucose, and high cholesterol [11]. These continue to be significant public health concerns in a number of developing countries. Hypertension, diabetes, tobacco use, dyslipidemia, overweight, and obesity have all been identified as a major risk factors for NCDs by researchers [8]. Taking into account the variations in outcome and exposure indicators, the burden of behavioral risk factors for NCDs is modified by socioeconomic features in resource-constrained

settings [12]. Socioeconomic status has been established as a significant predictor of the distribution of NCD risk factors [13] which has resulted in an increased interest in measuring health care inequality [14]. Moreover, inadequate resources, an aging population, and a weak health system all pose significant impediments to eliminate the burden of NCDs [13]. The general interest in socioeconomic inequality in health extends beyond their quantification to comprehending and analyzing their underlying causes [15].

Despite efforts to enhance people's lives and well-being, the burden of hypertension, diabetes, and other metabolic risk factors for NCDs continues to grow, with vast portions of the Bangladeshi population pleading for immediate action [4, 5, 16–19]. Hypertension and diabetes are the major risk factors for cardiovascular disease(CVDs), afflicting approximately 1 billion individuals worldwide living with hypertension [15]. According to the International Diabetes Federation (IDF), 693 million adults worldwide are expected to have diabetes by 2045 if effective preventative efforts are not put into place. Approximately 80% of those with diabetes live in low- and middle-income countries [20, 21]. Studies also indicate that Asia and the Eastern Pacific region will experience the hardest hit by this disease by 2045 [22]. According to the World Health Organization (WHO), 8% of Bangladeshi adults had diabetes in 2016, accounting for approximately 3% of all deaths in the country [23]. However, a recent nationwide survey revealed an upsurge of the prevalence of (diabetes 10%) [18] and (hypertension 24%) [4] in males and women in Bangladesh [24]. Economic development in general may result in an increase in sedentary lifestyle or a decrease in physical activity levels as a result of demographic transition and increased life expectancy. It may also result in the adoption of Western lifestyles, resulting in a nutritional shift toward unhealthy food choices, such as increased intake of "fast foods" high in sugar and fat may contribute to increase the diabetes and hypertension prevalence in Bangladesh [3–5, 25].

Limited resources, substandard public health system, an uncontrolled private health sector, and an aging population all pose substantial barriers to effectively addressing Bangladesh's escalating NCDs burden [13, 26]. In comparison to the plethora of evidence from high-income countries, there are comparatively few research on the link between socioeconomic position and NCDs in LMICs like Bangladesh. Previous studies demonstrated a disproportionate burden of several NCDs among rural low-income quintile populations and wealthier urban populations. The current study hypothesized that there is a significant variation in the distribution of contributing factors for NCDs by socioeconomic status. Based on the published literature, this study broke down inequality into a set of potential factors to determine their relative contributions. Therefore, this study attempts to measure the inequality of the prevalence of hypertension and diabetes, as well as identify factors that may contribute to socioeconomic inequalities in Bangladesh, using credible data. In addition, this study deconstructed inequality into a set of potential factors based on the existing literature in order to determine their respective contributions.

## Methods

### Data sources

The most recent Bangladesh Demographic and Health Survey (BDHS) dataset, 2017–18, was investigated in the study. The dataset was availed with appropriate application using DHS website. Between October 2017 and March 2018, the National Institute of Population Research and Training, Medical Education and Family Welfare Division, and the Ministry of Health and Family Welfare conducted the survey. The survey's primary goal was to measure health indicators and offer an overview of population, maternal, and child health, as well as the status of several noncommunicable diseases (NCDs) such as hypertension and diabetes.

## Study population and survey design

The 2017–18 BDHS used the complete list of enumeration areas (EAs) covering the total population of Bangladesh as its sampling frame. The survey used a list of enumeration areas (EAs) from the 2011 Population and Housing Census of the People's Republic of Bangladesh given by the Bangladesh Bureau of Statistics. The primary sampling unit (PSU) of the survey is an EA that covers an average of 120 households in 2017–18. The 2017–18 BDHS was a multistage stratified cluster sample of household's survey conducted in rural and urban settings, respectively. Rural wards were selected first, followed by PSUs, and then families were selected from PSUs. In urban areas, wards were selected using the PSUs technique, and each PSU had one EA. Then, households were picked from the sample of selected EAs. The final summary report of the 2017–18 BDHS contains a full discussion of the survey's design, procedures, sample size, questionnaires, and conclusions. Anthropometry and BP were also systematically measured from the selected subsample of 2017–18 BDHS [24]. A total of 12, 290 sample and who were not pregnant considered in the analysis.

## Outcome variables

Two outcome variables hypertension and diabetes were used for the analysis. Trained health technicians measured BP three times using digital oscillometric blood pressure measuring device monitor at about five minutes interval [24]. Then, the average of second and third measurements was used to report respondents' final BP [24]. The American College of Cardiology/ American Heart Association (ACC/AHA-2017) guidelines, individuals with a Systolic Blood Pressure (SBP of ≥130mmHg) and/or a Diastolic Blood Pressure (DBP of ≥80mmHg) or who were taking any prescribed antihypertensive drugs to control BP were categorized as hypertensive [27]. The Fasting Blood Glucose (FPG) (≥7.0) mmol/l (126 mg/dl) was classified as diabetes [28]. In addition, participants who were taking prescribed medications to lower their elevated blood glucose levels were classified as having diabetes [24].

## Independent variables

The independent variables included in the study were selected based on previous literature reporting the risk of hypertension and diabetes in LMICs setting [4, 5, 8, 13, 17, 19, 29–32]. The household factors included administrative divisions (Barisal, Chittagong, Dhaka, Khulna, Rajshahi, Rangpur, Sylhet, Mymensingh); place of residence (urban, rural); and wealth status (poorest, poorer, middle, richer, richest) based on pre-set cutoffs, whereas the socioeconomic and individual factors included: age of the participants (18–24, 25–34, 35–49, 50–59, 60–69, ≥70); sex of the participants (men, women); education level (no education, primary, secondary, higher); and occupational status (had no work, had work) and body mass index (BMI) level. We have used global cut-off points for BMI classification: underweight ($<18.5$kg/m$^2$), normal (18.5–25.0kg/m$^2$), overweight (25.1–29.9kg/m$^2$), and obese (≥30.0 kg/m$^2$) [33].

**Conceptual framework.** The distribution of different predictor variables and their combined effects have been shown in the (**Fig 1**). Despite strong associations between predictors and burden of NCDs (hypertension and diabetes) are separate entities, and each has an independent effect on outcomes [34]. To investigate the effects, it was hypothesized that several predictors (e.g., individual demographic factors, socioeconomic factors, geographical factors, lifestyle factors and intermediary factors BMI) were associated with developing outcomes (hypertension and diabetes) aligned with previously did studies elsewhere [3, 4, 5, 15–18, 30, 31, 35–46,] and the combination of predictors was expected to predict people's health outcomes (hypertension and diabetes) [34].

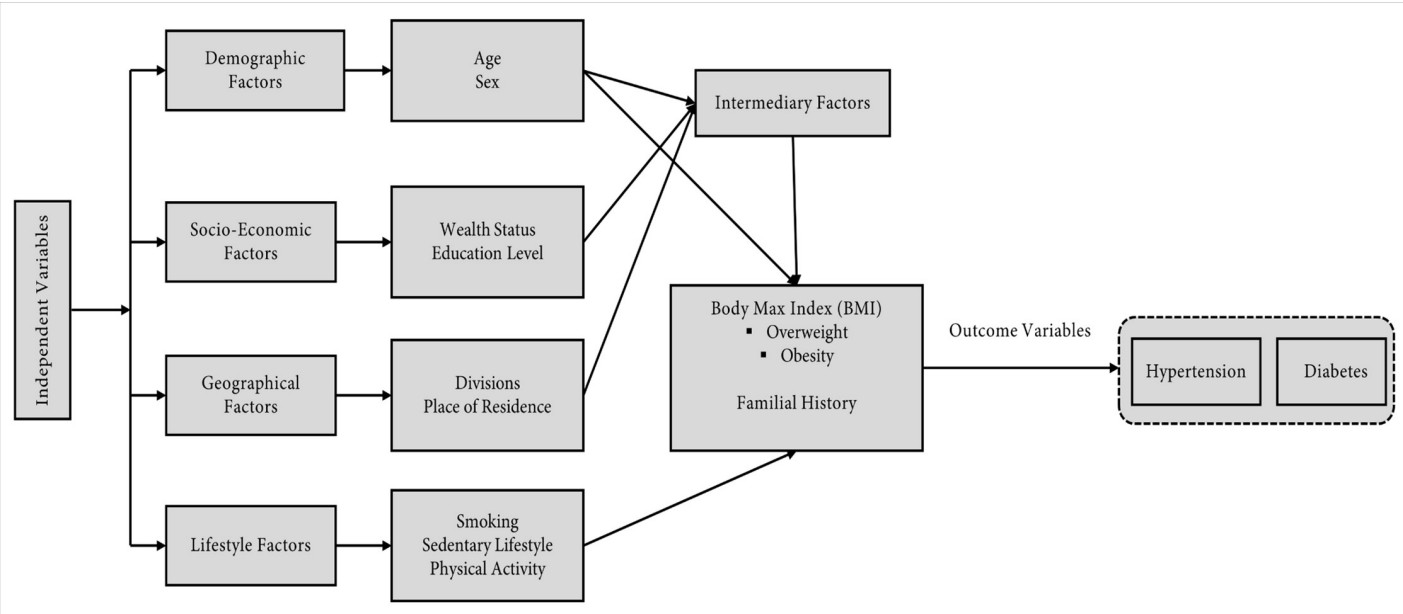

**Fig 1. Conceptual framework of the study.**

## Statistical analyses

Given the complexity of the BDHS survey, the data were pre-processed and weighted using savy command. Following that, the paper examined the normalcy assumption for continuous variables based on their distribution and reported it using medians and interquartile ranges (IQRs). Then, hypertension and diabetes were measured using age-standardised methods. As a result, the prevalence of hypertension and diabetes were normalized in the same standard population in order to eliminate or reduce the effect of participant age and sex distribution discrepancies. The selected explanatory variables were then fitted in an unadjusted log-binomial regression model. Following that, a log-binomial passion regression model with survey weights and explanatory variables with p-values (0.05) was used to identify risk factors for hypertension and diabetes, presenting results as adjusted incidence rate ratio(aIRR) and their 95% confidence intervals (CIs) and p-values. For data analysis, this study used Stata/MP 16 (StataCorp, College Station, Texas, USA).

## Inequality measurement

The effects and contributions of various socioeconomic and demographic aspects of respondents to wealth-related disparity are shown using decomposition analysis. This technique can offer information on how hypertension and diabetes respond to changes in the determinant's variable, which is important for prioritizing public health measures [18, 47]. To accomplish the objective of this study, the analyses were performed in several stages: plotting the concentration curves, examining the concentration indexes (CIs), and decomposition of the concentration index. To begin, we used concentration curves to look at how hypertension and diabetes prevalence differed by socioeconomic status. The cumulative percentage of respondents with hypertension and diabetes is plotted against the cumulative percentage of respondents according to their socioeconomic status on a concentration curve, which measures inequality (wealth index). The 45˚ line represents the equality line, indicating that there is no socioeconomic inequality in hypertension and diabetes. The presence of hypertension and

diabetes is more concentrated among impoverished respondents if the concentration curve is above the equality line. If the concentration curve falls below the equality line, it means that the prevalence of hypertension and diabetes is unequally distributed among the upper (wealthy) class.

The concentration curve (CC) and concentration index (CIX) in their relative formulation (with no correction), were used to investigate the inequality in terms of hypertension and diabetes across analyzable socio-economic characteristics of the population [48]. The CIX in this study represents horizontal inequity, as each individual in the study was presumed to have an equal requirement. While constructing CC, the cumulative proportion of individual ranked according to their wealth index score (poorest first) was plotted against the cumulative proportion of hypertension and diabetes on the y-axis. The 45-degree slope from the origin revealed perfect equality. If the CC overlaps with the line of equality, hypertension and diabetes are equal among the population. However, if the CC subtends the line of equality below (above), then inequality in the use of hypertension and diabetes exists and is slanted towards individual belonging to low (high) socio-economic background. Further the CC subtends from the line of equality, the greater the degree of inequality. To evaluate the extent of wealth-related inequality, CIX was determined. CIX is widened as twice the region between the line of equality and CC [48].

The following are the advantages of using CIX as a measure of inequality index in healthcare: it takes socioeconomic dimension of hypertension and diabetes into account since the classification of individuals is according to the socioeconomic status, instead of their health status; it captures the experience of the whole population and; it is sensitive towards the changes in population distribution across socioeconomic groups. The CIX takes a value between − 1 and + 1. When the hypertension and diabetes are equally distributed across socioeconomic groups, CIX takes the value of 0. A positive value of CIX implies that hypertension and diabetes are concentrated among the higher socioeconomic groups (pro-rich). Conversely, a negative value of CIX suggests that hypertension and diabetes are concentrated among the lower socioeconomic groups (pro-poor). The calculation of CIX was done by using "convenient covariance" formula described by O'Donnell et al. [48], as shown in the Eq. 1 below.

$$CIX = \frac{2}{\mu} cov\,(h, r) \tag{1}$$

Here $h$ is the health sector variable, $\mu$ is its mean, and $r = i/N$ is the fractional rank of individual $i$ in the living standards distribution, with $i = 1$ for the poorest and $i = N$ for the richest. The user-written STATA commands Lorenz [49] and conindex [50] were used to produce CC and measure CIX, respectively.

Finally, the relative CIX was decomposed to determine the portion of inequality owing to the inequality in the underlying determinants. The findings were analyzed and interpreted using the technique described by Wagstaff et al. [51] and O'Donnell et al. [52]. The contribution of each determinant to overall wealth-related inequality is determined as the product of the determinant's sensitivity to hypertension and diabetes (elasticity) and the degree of wealth-related inequality in that determinant (CIX of determinant). The residual is the portion of the CIX that is not explained by the determinants. The results of the decomposition method were reported in the following formats: elasticity, concentration index value, absolute contribution (same unit as the concentration index), and percentage (relative) contribution.

The "elasticity" column refers to the change in the dependent variable (socioeconomic inequality in hypertension and diabetes) that results from a one-unit change in the explanatory variables. A positive or negative sign in the elasticity denotes a rising or falling trend in hypertension and diabetes in association with a positive change in the determinants [47]. The

distribution of the determinants in relation to the wealth quintiles is shown in the column "CI." The positive or negative sign of "CI" denotes whether the components were more concentrated in rich or poor households, respectively [18]. The percentage contribution indicates the relative contribution of each model component to the overall socioeconomic inequality in hypertension and diabetes. The observed socioeconomic inequality of hypertension and diabetes is increased by factors with positive percentage contributions. A negative percentage contribution, on the other hand, denotes a factor that is expected to lower the observed socioeconomic inequality in hypertension and diabetes [18].

## Results

### Background characteristics of study objects

A total of 12,290 were included in the final analysis, with 57.18% of them being women, 29.92% having primary education, 58.65% having a normal BMI, and 23.08% from Dhaka division. Approximately 73.47% lived in rural areas, and 32.71% were between the years of age 18–24 bracket (**Table 1**).

### Prevalence of diabetes and hypertension

**Table 2** shows the age-standardized prevalence of having hypertension and diabetes among the sample population. The overall age-standardized prevalence of hypertension found (36.98% 95% CI: 36.79–37.16) and diabetes (11.29% 95% CI: 11.13–11.69). Diabetes was found to be somewhat more prevalent in women aged 18 or older (9.96% 95% CI: 9.83–10.08). The prevalence of diabetes and hypertension are substantially more prevalent among urban women (15.69% 95% CI: 15.45–15.93) and (38.65% 95% CI: 38.34–38.97) comparing to rural women. Diabetes prevalence was significantly higher among individuals aged 50–59 years. (17.85% 95% CI: 17.47–18.23) while hypertension found higher among the age 70+ or older. The diabetes and hypertension prevalence were higher among overweight and obesity BMI groups. Diabetes was found to be slightly more prominent among those who had completed secondary education. (11.05% 95% CI: 10.87–11.23) while hypertension found higher among the people who had no education (58.0% 95% CI: 56.2–59.7). The hypertension and diabetes prevalence increased with respondents' wealth status. Highest prevalence of diabetes found among the two top wealth quantiles richest (20.27% 95% CI: 19.94–20.59) and richer (13.23% 95% CI: 12.94–13.52). While the hypertension found more common in richest group (41.71% 95% CI: 41.31–42.11). Across the divisions, diabetes found higher among the city dwellers of Dhaka (16.46% 95% CI: 16.06–16.87) and lowest in (7.00% 95% CI: 6.73–7.27).

**Table 3**, reports the adjusted and unadjusted log binomial passion distribution results. To interpret the findings researcher only discuss the adjusted incidence rate ratio (aIRR). The hypertension rate among the women individuals 1.13 times (aIRR: 1.13, 95% CI: 1.06–1.20; <0.001) high comparing to males. With age, the rate of hypertension increased while highest rate observed age ≥70 years or older 6.23 times (aIRR: 6.23, 95% CI: 5.25–7.03; <0.001). Diabetes rates have risen in line with the same age-related pattern. The rate of hypertension is 2.37–2.41 times higher in those with a BMI of 25.1–29.9 or ≥30 than in those with a BMI of less than 25. People with a BMI of 25.1–29.9—or 30—had 2.00- and 2.22-times risk of diabetes, respectively, as do those with a BMI of less than 25. The aIRR was much greater among individuals with some level of formal education. Individuals in the middle, upper, or uppermost quintiles of wealth had a greater aIRR than those in the poorest quintile for both hypertension and diabetes. In comparison to Dhaka, all of the surveyed divisions had lower aIRR for diabetes however, dissimilarities found in the hypertension rate. Alternatively, people who were

**Table 1. General characteristics of the analyzable objects (unweighted).**

| Administrative Divisions | | Total (weighted) | Poorest | Poorer | Middle | Richer | Richest |
|---|---|---|---|---|---|---|---|
| | Barisal | 666(5.51) | 371(29.1) | 282(22.05) | 246(19.23) | 200(15.64) | 180(14.07) |
| | Chittagong | 2081(17.21) | 187(11.23) | 275(16.52) | 337(20.24) | 328(19.70) | 538(32.31) |
| | Dhaka | 2790(23.08) | 145(9.05) | 170(10.61) | 244(15.22) | 441(27.51) | 603(37.62) |
| | Khulna | 1511(12.50) | 164(9.62) | 327(19.18) | 402(23.58) | 411(24.11) | 401(23.52) |
| | Mymensingh | 988(8.17) | 384(27.53) | 321(23.01) | 300(21.51) | 206(14.77) | 184(13.19) |
| | Rajshahi | 1751(14.48) | 290(17.98) | 347(21.51) | 398(24.67) | 338(20.95) | 240(14.88) |
| | Rangpur | 1514(12.53) | 556(35.17) | 350(22.14) | 271(17.14) | 198(12.52) | 206(13.03) |
| | Sylhet | 788(6.52) | 294(20.29) | 263(18.15) | 234(16.15) | 294(20.29) | 364(25.12) |
| **Place of Residence** | | | | | | | |
| | Urban | 3207(26.53) | 382(8.71) | 376(8.57) | 667(15.20) | 1120(25.52) | 1843(42.00) |
| | Rural | 8884(73.47) | 2009(25.42) | 1959(24.79) | 1765(22.34) | 1296(14.40) | 873(11.05) |
| **Sex of the Respondents** | | | | | | | |
| | Male | 5178(42.82) | 1002(18.93) | 1009(19.06) | 1060(20.03) | 1059(20.01) | 1163(21.97) |
| | Women | 6914(57.18) | 1389(19.85) | 1326(18.95) | 1372(19.61) | 1357(19.39) | 1553(22.20) |
| **Age of the Participants (years)** | | | | | | | |
| Median (IQR): 31.0 (23.0–43.0) | | | | | | | |
| | 18–24 | 3955(32.71) | 733(18.38) | 740(18.56) | 787(19.74) | 832(20.87) | 895(22.45) |
| | 25–34 | 2841(23.50) | 575(19.84) | 535(18.46) | 568(19.60) | 544(18.77) | 676(23.33) |
| | 35–49 | 2052(16.97) | 400(19.17) | 412(19.74) | 410(19.65) | 420(20.12) | 445(21.32) |
| | 50–59 | 1360(11.25) | 260(18.77) | 265(19.13) | 285(20.58) | 252(18.19) | 323(23.32) |
| | 60–69 | 1062(8.79) | 245(22.09) | 225(20.29) | 204(18.39) | 218(19.66) | 217(19.57) |
| | ≥70 | 818(6.77) | 178(21.60) | 158(19.17) | 178(21.60) | 150(18.20) | 160(19.42) |
| **BMI Level** | | | | | | | |
| Median (IQR): 21.44 (19.13–24.36) | | | | | | | |
| | Underweight (<18.5 kg/m$^2$) | 2066(17.28) | 652(31.35) | 517(24.86) | 410(19.71) | 302(14.52) | 199(9.57) |
| | Normal (18.5–25.0 kg/m$^2$) | 7014(58.65) | 1450(20.37) | 1429(20.08) | 1498(21.05) | 1452(20.40) | 1288(18.10) |
| | Overweight (25.1–29.9 kg/m$^2$) | 2388(19.97) | 246(10.05) | 321(13.11) | 429(17.52) | 536(21.90) | 916(37.42) |
| | Obesity (≥30.0 kg/m$^2$) | 489(4.09) | 18(3.55) | 33(6.51) | 71(14.00) | 96(18.93) | 289(57.00) |
| **Education Level** | | | | | | | |
| | No Education | 3114(25.75) | 973(31.95) | 738(24.24) | 604(19.84) | 446(14.65) | 284(9.33) |
| | Primary | 3617(29.92) | 913(24.68) | 883(23.86) | 776(20.97) | 644(17.41) | 484(13.08) |
| | Secondary Education | 3561(29.46) | 420(11.87) | 583(16.47) | 751(21.22) | 869(24.55) | 916(25.88) |
| | Higher | 1798(14.87) | 85(4.24) | 131(6.53) | 301(15.00) | 457(22.78) | 1032(51.45) |
| **Working Status** | | | | | | | |
| | Had no Work | 4372(39.13) | 713(14.70) | 757(15.61) | 938(19.34) | 1028(21.20) | 1413(29.14) |
| | Had Work | 7360(60.87) | 1678(22.55) | 1578(21.21) | 1494(20.08) | 1388(18.65) | 1303(17.51) |

BMI stands for Body Mass Index, IQR for Interquartile Range

engaged in any kinds of income-generating activities decreased the risk of having hypertension and diabetes.

## Inequality in hypertension and diabetes

The concentration curve and concentration index were used to estimate and to show the inequalities in hypertension and diabetes among Bangladeshi individuals (**Figs 2 & 3**). In both situations, the concentration curve is located below the line of equality (45˚ line), indicating a

**Table 2. Age-standardized prevalence of diabetes and hypertension with corresponding 95% confidence interval (95% CI).**

|  |  | Diabetes | Hypertension |
|---|---|---|---|
| **Overall Prevalence** |  | 11.57(11.45–11.69) | 36.98(36.79–37.16) |
| **Administrative Divisions** |  |  |  |
|  | Barisal | 11.45(11.09–11.82) | 42.00(41.44–42.57) |
|  | Chittagong | 13.58(13.23–13.93) | 39.49(39.00–39.99) |
|  | Dhaka | 16.46(16.06–16.87) | 31.13(30.62–31.63) |
|  | Khulna | 11.54(11.22–11.86) | 39.93(39.43–40.42) |
|  | Mymensingh | 10.04(9.72–10.38) | 33.23(32.72–33.75) |
|  | Rajshahi | 9.82(9.51–10.14) | 35.50(35.00–36.00) |
|  | Rangpur | 7.00(6.73–7.27) | 38.95(38.43–39.46) |
|  | Sylhet | 12.75(12.38–13.13) | 35.05(34.52–35.58) |
| **Place of Residence** |  |  |  |
|  | Urban | 15.69(15.45–15.93) | 38.65(38.34–38.97) |
|  | Rural | 9.46(9.33–9.6) | 36.12(35.9–36.35) |
| **Sex of the Respondents** |  |  |  |
|  | Men | 9.71 (9.60–9.82) | 26.34(25.87–27.41) |
|  | Women | 9.96 (9.83–10.08) | 28.61 (27.3–29.31) |
| **Age of the Participants (years)** |  |  |  |
|  | 18–24 (RC) | 4.43(4.27–4.6) | 9.68(9.45–9.92) |
|  | 25–34 | 9.9(9.65–10.14) | 26.33(25.97–26.69) |
|  | 35–49 | 13.43(13.14–13.73) | 41.66(41.24–42.09) |
|  | 50–59 | 17.85(17.47–18.23) | 50.31(49.81–50.81) |
|  | 60–69 | 14.59(14.22–14.98) | 57.68(57.14–58.21) |
|  | ≥70 | 14.22(13.83–14.62) | 63.35(62.8–63.9) |
| **BMI Level** |  |  |  |
|  | Underweight ($<18.5 \text{ kg/m}^2$) | 6.42(6.19–6.66) | 28.41(27.99–28.84) |
|  | Normal (18.5–25.0 $\text{kg/m}^2$) | 9.52(9.37–9.67) | 32.89(32.65–33.14) |
|  | Overweight (25.1–29.9 $\text{kg/m}^2$) | 16.84(16.54–17.13) | 46.83(46.44–47.22) |
|  | Obesity (≥30.0 $\text{kg/m}^2$) | 20.35(19.73–20.98) | 51.41(50.63–52.18) |
| **Education Level** |  |  |  |
|  | No Education | 8.49 (8.35–8.63) | 58.0 (56.2–59.7) |
|  | Primary | 9.80 (9.65–9.96) | 49.5 (47.9–51.1) |
|  | Secondary Education | 11.05 (10.87–11.23) | 48.4 (46.7–50.0) |
|  | Higher | 10.79 (10.55–11.04) | 50.7 (48.5–52.9) |
| **Working Status** |  |  |  |
|  | Had no Work | 13.72(13.56–13.89) | 41.57(41.33–41.81) |
|  | Had Work | 8.37(8.21–8.54) | 30.19(29.92–30.46) |
| **Wealth status** |  |  |  |
|  | Poorest | 7.66(7.44–7.89) | 33.47(33.07–33.86) |
|  | Poorer | 6.6(6.39–6.82) | 34.09(33.68–34.5) |
|  | Middle | 9.14(8.9–9.39) | 36.86(36.45–37.28) |
|  | Richer | 13.23(12.94–13.52) | 38.31(37.89–38.73) |
|  | Richest | 20.27(19.94–20.59) | 41.71(41.31–42.11) |

larger concentration of hypertension and among individuals from the top wealth quintiles than among the rest of the population. This study revealed that the values of the CIX for hypertension and diabetes were (CIX: 0.06388445 (p<0.001) and (CIX: 0.24727198 (p<0.001),

**Table 3. Factors associated with hypertension and diabetes: BDHS 2017–18.**

| Administrative Divisions | Hypertension | | | | Diabetes | | | |
|---|---|---|---|---|---|---|---|---|
| | UIRR (95% CI) | P-Value | aIRR (95% CI) | P-Value | UIRR (95% CI) | P-Value | aIRR (95% CI) | P-Value |
| Dhaka (RC) | 1 | | 1 | | 1 | | 1 | |
| Barisal | 1.34(1.20–1.51) | <0.001 | 1.28(1.45–1.43) | <0.001 | 0.68(0.56–0.83) | <0.001 | 0.77(0.63–0.94) | 0.011 |
| Chittagong | 1.26(1.13–1.41) | <0.001 | 1.21(1.09–1.35) | <0.001 | 0.76(0.63–0.91) | 0.002 | 0.75(0.63–0.90) | 0.001 |
| Khulna | 1.27(1.14–1.42) | <0.001 | 1.16(1.05–1.29) | 0.004 | 0.61(0.50–0.74) | <0.001 | 0.60(0.50–0.72) | <0.001 |
| Mymensingh | 0.99(0.87–1.13) | 0.894 | 1.03(0.92–1.17) | 0.590 | 0.54(0.44–0.67) | <0.001 | 0.68(0.55–0.85) | 0.001 |
| Rajshahi | 1.16(1.03–1.30) | 0.015 | 1.18(1.07–1.32) | 0.002 | 0.58(0.48–0.70) | <0.001 | 0.68(0.56–0.83) | <0.001 |
| Rangpur | 1.29(1.16–1.45) | <0.001 | 1.37(1.23–1.53) | <0.001 | 0.41(0.33–0.52) | <0.001 | 0.52(0.42–0.66) | <0.001 |
| Sylhet | 1.08(0.96–1.22) | 0.217 | 1.18(1.05–1.32) | 0.004 | 0.67(0.55–0.81) | <0.001 | 0.75(0.62–0.91) | 0.003 |
| **Place of Residence** | | | | | | | | |
| Rural (RC) | 1 | | | | 1 | | 1 | |
| Urban | 1.08(1.02–1.15) | <0.001 | 1.01(0.95–1.07) | 0.739 | 1.44(1.29–1.60) | <0.001 | 0.99(0.88–1.12) | 0.932 |
| **Sex of the Respondents** | | | | | | | | |
| Male (RC) | 1 | | 1 | | 1 | | 1 | |
| Women | 1.08(1.2–1.14) | 0.010 | 1.13(1.06–1.20) | <0.001 | 0.98(0.88–1.10) | 0.784 | 0.95(0.83–1.09) | 0.489 |
| **Age of the Participants (years)** | | | | | | | | |
| 18–24 (RC) | 1 | | 1 | | 1 | | 1 | |
| 25–34 | 2.26(2.02–2.53) | <0.001 | 2.11(1.88–2.37) | <0.001 | 2.05(1.70–2.47) | <0.001 | 1.97(1.62–2.39) | <0.001 |
| 35–49 | 3.35(3.00–3.73) | <0.001 | 3.21(2.86–3.60) | <0.001 | 2.84(2.36–3.42) | <0.001 | 2.91(2.38–3.54) | <0.001 |
| 50–59 | 4.27(3.83–4.75) | <0.001 | 4.31(3.84–4.84) | <0.001 | 3.83(3.18–4.63) | <0.001 | 3.96(3.24–4.84) | <0.001 |
| 60–69 | 4.96(4.45–5.53) | <0.001 | 5.21(4.64–5.85) | <0.001 | 3.51(2.87–4.30) | <0.001 | 3.96(3.19–4.92) | <0.001 |
| ≥70 | 5.65(5.07-.29) | <0.001 | 6.23(5.25–7.03) | <0.001 | 3.46(2.78–4.31) | <0.001 | 3.85(3.01–4.93) | <0.001 |
| **BMI Level** | | | | | | | | |
| Underweight (<18.5 kg/m$^2$) (RC) | 1 | | 1 | | 1 | | 1 | |
| Normal (18.5–25.0 kg/m$^2$) | 1.35(1.22–1.45) | <0.001 | 1.46(1.32–1.60) | <0.001 | 1.52(1.25–1.85) | <0.001 | 1.41(1.16–1.71) | 0.001 |
| Overweight (25.1–29.9 kg/m$^2$) | 2.30(2.08–2.54) | <0.001 | 2.37(2.15–2.62) | <0.001 | 2.70(2.21–3.31) | <0.001 | 2.00(1.61–2.47) | <0.001 |
| Obesity (≥30.0 kg/m$^2$) | 2.49(2.19–2.84) | <0.001 | 2.41(2.11–2.75) | <0.001 | 3.57(2.78–4.58) | <0.001 | 2.22(1.71–2.89) | <0.001 |
| **Education Level** | | | | | | | | |
| No Education (RC) | 1 | | 1 | | 1 | | 1 | |
| Primary | 0.74(0.69–0.79) | <0.001 | 1.00(0.93–1.07) | 0.988 | 1.03(0.89–1.19) | 0.671 | 1.25(1.08–1.46) | 0.003 |
| Secondary Education | 0.65(0.60–0.70) | <0.001 | 1.01(0.93–1.09) | 0.826 | 1.02(0.88–1.18) | 0.823 | 1.21(1.02–1.44) | 0.027 |
| Higher | 0.66(0.60–0.72) | <0.001 | 1.06(0.96–1.17) | 0.245 | 1.06(0.90–1.26) | 0.473 | 1.13(0.92–1.38) | 0.251 |
| **Working Status** | | | | | | | | |
| Had no Work (RC) | 1 | | 1 | | 1 | | 1 | |
| Had Work | 0.79(0.75–0.84) | <0.001 | 0.92(0.86–0.98) | 0.010 | 0.73(0.66–0.82) | <0.001 | 0.83(0.72–0.95) | 0.007 |
| **Wealth Status** | | | | | | | | |
| Poorest (RC) | 1 | | 1 | | 1 | | 1 | |
| Poorer | 1.09(0.98–1.20) | 0.099 | 1.06(0.97–1.17) | 0.201 | 1.02(0.81–1.28) | 0.889 | 0.97(0.77–1.22) | 0.777 |
| Middle | 1.18(1.07–1.30) | 0.001 | 1.11(1.01–1.22) | 0.023 | 1.28(1.11–1.70) | 0.003 | 1.22(0.98–1.52) | 0.075 |
| Richer | 1.27(1.16–1.40) | <0.001 | 1.17(1.06–1.29) | 0.001 | 1.90(1.56–2.32) | <0.001 | 1.53(1.24–1.91) | <0.001 |
| Richest | 1.48(1.35–1.61) | <0.001 | 1.18(1.07–1.31) | 0.001 | 3.06(2.55–3.67) | <0.001 | 2.15(1.73–2.68) | <0.001 |

RC stands for Reference Category, UIRR means Unadjusted incidence rate ratio, AIRR means Adjusted incidence rate ratio

respectively, for hypertension and diabetes (**Figs 2 & 3**). Hence, among Bangladeshi households with greater socioeconomic standing, this study discovered a pro-rich socioeconomic inequality for hypertension and diabetes.

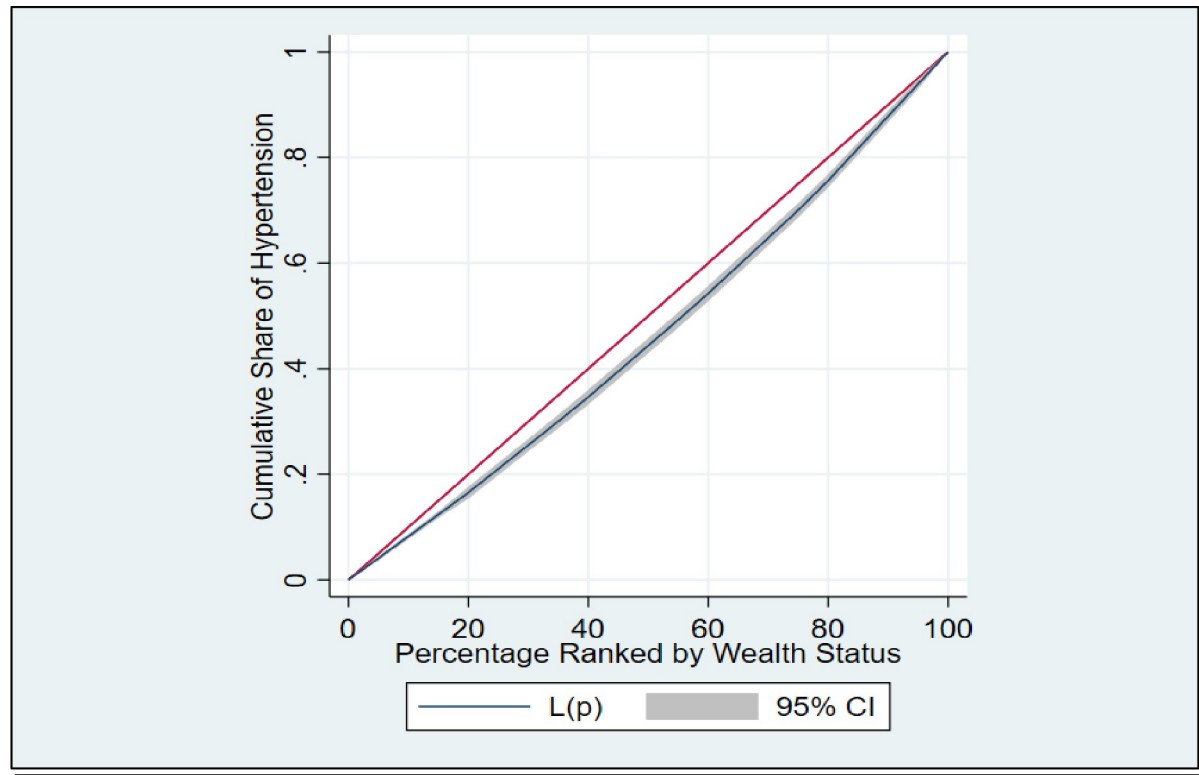

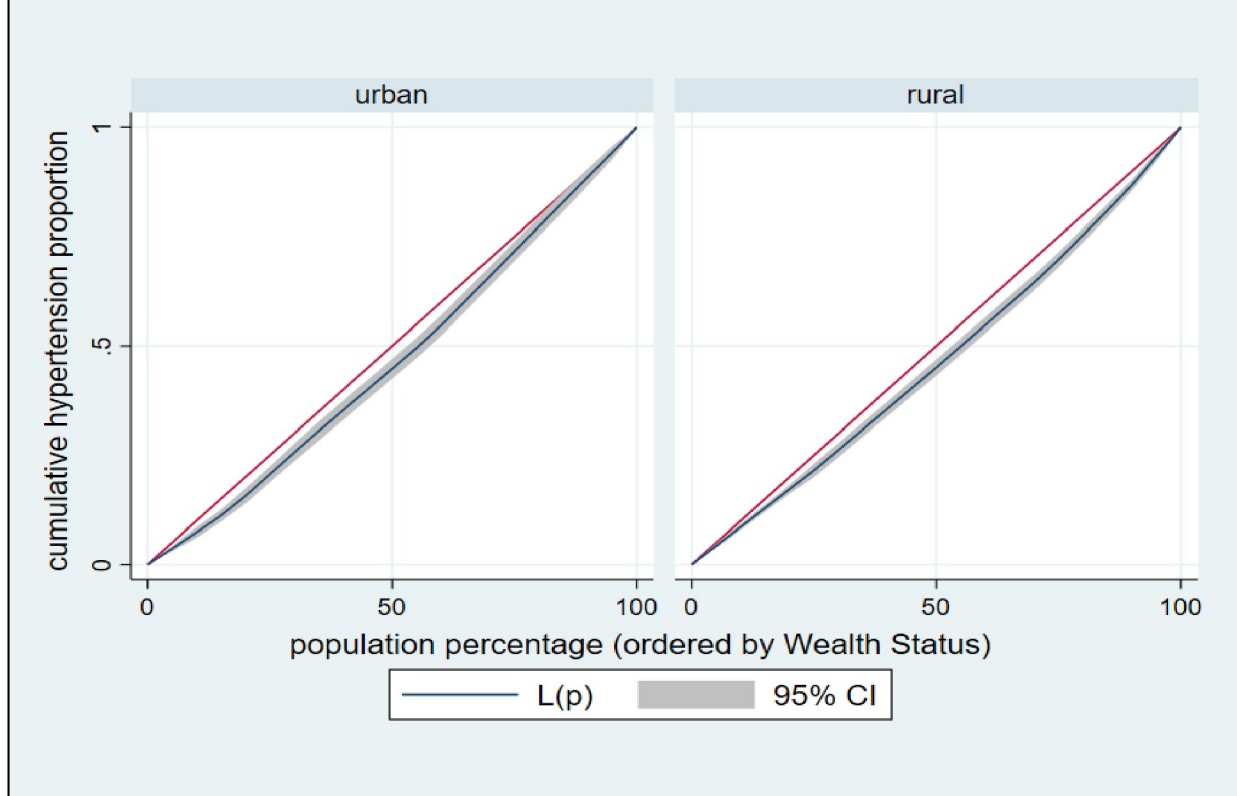

**Fig 2. Concentration curve for hypertension over wealth status.**

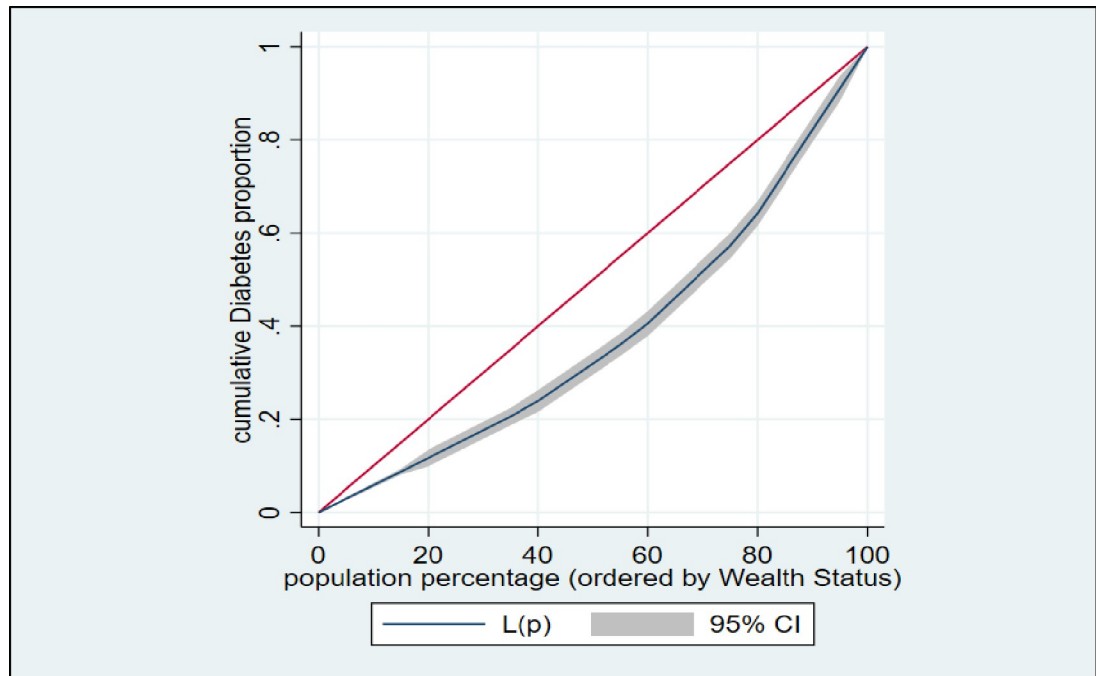

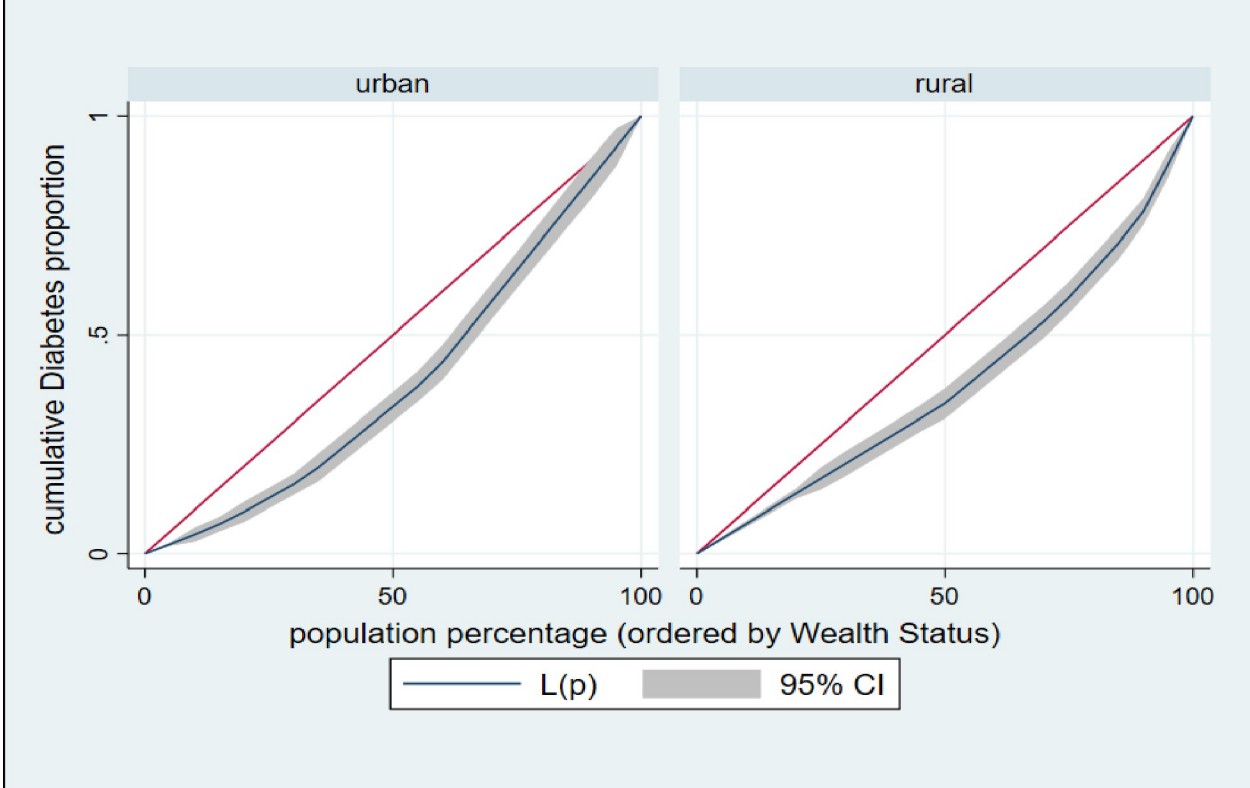

**Fig 3. Concentration curve for diabetes over wealth status.**

### Decomposition of concentration index for hypertension and diabetes

**Table 4** demonstrates the effects of several socioeconomic and demographic determinants on hypertension and diabetes inequities. The column 'Elasticity' indicates the amount of change in the dependent variable (socioeconomic inequality in hypertension and diabetes) that occurs when the explanatory factors change by one unit. Elasticity with a positive or negative sign implies an increasing or decreasing trend in diabetes or hypertension in conjunction with a positive change in the factor [18, 47]. The distribution of the determinants in terms of wealth quintiles is shown by the column 'CIX.' The positive or negative direction of the CI signifies that the factors were more concentrated in either wealthy or impoverished groups. The percentage contribution illustrates how much each determinant in the model contributes to overall socioeconomic inequalities. A positive percentage contribution means a factor contributes to increase observed socioeconomic disparities of hypertension and diabetes. A negative percentage contribution, on the other hand, denotes a factor that is expected to reduce hypertension and diabetes-related socioeconomic inequalities. In Bangladesh, household wealth status accounted for roughly 25.71% an 43.41% of the overall inequality in hypertension and diabetes, respectively. While BMI made a significant impact to the establishment of inequality concentration, it was 4.95 and 83.38% for hypertension and diabetes, correspondingly. Furthermore, urban areas were responsible for 4.56% of the intensification in diabetes among Bangladeshi residents, while administrative regions were responsible for 4.76% of the hypertension inequality. While educational status accounts for 7.23% and 0.89% of hypertension and diabetes inequality in Bangladeshi adults, respectively. The unexplained or residual contributing factors to the socioeconomic inequalities in hypertension and diabetes accounted for (-38. 58%) and (-62.14%), respectively.

## Discussion

The current study investigated the socioeconomic inequalities associated with metabolic and behavioural risk factors for hypertension and diabetes in the Bangladeshi population using the most recent demographic and health survey data. Socioeconomic inequality analysis has developed into a critical instrument for influencing policy decisions that are driven by inequities. The analysis found that diabetes and hypertension are more common and concentrated among the richest Bangladeshis living in urban areas, and their incidence is noticeably higher than in the last round of research. The pro-rich socioeconomic inequalities in hypertension and diabetes were significantly influenced by household wealth status, administrative region, being overweight or obese, employment status, and living in metropolitan areas. A report published in 2013 elsewhere documented that persons who live in impoverished or disadvantaged communities face a greater risk of death from non-communicable diseases than those who reside in more advantaged groups and communities [53]. In eleven European countries, a study found that ischemic heart disease is more prevalent in lower socioeconomic groups, in contrast to India, where cardiovascular illness and cardio-metabolic risk factors are more prevalent in higher socioeconomic groups [36, 54, 55]. Although another study in Southeast Asia found that a disproportionate number of unfavourable risk factors for NCDs are concentrated in impoverished groups [37]. According to a study comprising 41 low- and middle-income nations, wealth and education were found to be inversely connected to numerous NCDs, and poorer socioeconomic position may be responsible for raising the risk of NCD mortality [56, 57].

The age-adjusted prevalence of hypertension and diabetes are significantly lower among the people from less socioeconomic status than higher socioeconomic status (**see Table 3**), which is consistent with earlier studies elsewhere [4, 5, 15]. Economic development, in general,

**Table 4. Decomposition of concentration index for measuring socioeconomic inequalities in hypertension and diabetes.**

| | Hypertension | | | | Diabetes | | | |
|---|---|---|---|---|---|---|---|---|
| | | | Contribution to overall CIX = 0.06388445 (p<0.001) | | | | Contribution to overall CIX = 0.24727198 (p<0.001) | |
| | Elasticity | CIX | Absolute contribution | Percentage contribution | Elasticity | CIX | Absolute contribution | Percentage contribution |
| **Administrative Divisions** | | | | | | | | |
| Dhaka (RC) | | | | | | | | |
| Barisal | .01449863 | -.23288488 | -.00337651 | -5.2853415 | -.00551451 | -.23288488 | .00128425 | .51936546 |
| Chittagong | .03011325 | .12439611 | .00374597 | 5.8636667 | -.0172004 | .12439611 | -.00213966 | -.86530734 |
| Khulna | .01850648 | .05179982 | .00095863 | 1.500572 | -.02543476 | .05179982 | -.00131752 | -.5328206 |
| Mymensingh | .00314632 | -.21009727 | -.00066103 | -1.0347312 | -.01129641 | -.21009727 | .00237335 | .95981164 |
| Rajshahi | .02980763 | -.09989361 | -.00297759 | -4.6609016 | -.02127521 | -.09989361 | .00212526 | .85948189 |
| Rangpur | .0430925 | -.29738631 | -.01281512 | -20.059844 | -.03104046 | -.29738631 | .00923101 | 3.7331395 |
| Sylhet | .01269142 | -.02917272 | -.00037024 | -.57955149 | -.0070774 | -.02917272 | .00020647 | .08349791 |
| Subtotal | | | | -24.25613109 | | | 0.01176316 | 4.75716846 |
| **Place of Residence** | | | | | | | | |
| Urban | -.0213918 | -.13629789 | .00291566 | **4.5639544** | .0010518 | -.13629789 | -.00014336 | -.05797604 |
| Rural (RC) | | | | | | | | |
| **Sex of the Respondents** | | | | | | | | |
| Male (RC) | | | | | | | | |
| Women | -.05691692 | .00645033 | -.00036713 | -.57468238 | .02431801 | .00645033 | .00015686 | .06343585 |
| **Age of the Participants (years)** | | | | | | | | |
| 18–24 (RC) | | | | | | | | |
| 25–34 | .13745838 | .00559709 | .00076937 | 1.2043098 | .0594244 | .00559709 | .0003326 | .13450922 |
| 35–49 | .16758381 | -.00867759 | -.00145422 | -2.2763347 | .0720604 | -.00867759 | -.00062531 | -.25288378 |
| 50–59 | .15242511 | -.00836368 | -.00127483 | -1.995532 | .05990346 | -.00836368 | -.00050101 | -.20261629 |
| 60–69 | .14061338 | -.05352401 | -.00752619 | -11.780945 | .04830372 | -.05352401 | -.00258541 | -1.0455729 |
| ≥70 | .12281228 | -.03691302 | -.00453337 | -7.0962063 | .03402716 | -.03691302 | -.00125605 | -.507961 |
| Subtotal | | | -0.01401924 | **-21.9447082** | | | -0.00463518 | **-1.87452475** |
| **BMI Level** | | | | | | | | |
| Underweight (<18.5 kg/m²) (RC) | | | | | | | | |
| Normal (18.5–25.0 kg/m²) | .23974132 | -.03916543 | -.00938957 | -14.697743 | .05802614 | -.03916543 | -.00227262 | -.91907665 |
| Overweight (25.1–29.9 kg/m²) | .19997692 | .22286179 | .22286179 | 69.762229 | .04395601 | .22286179 | .00979611 | 3.9616758 |
| Obesity (≥30.0 kg/m²) | .04159134 | .43494286 | .01808986 | 28.316526 | .01086356 | .43494286 | .00472503 | 1.910862 |
| Subtotal | | | 0.23156208 | 83.381012 | | | 0.01224852 | 4.95346115 |
| **Education Level** | | | | | | | | |
| No Education (RC) | | | | | | | | |
| Primary | .00031321 | -.13044191 | -.00004086 | -.06395154 | .03157589 | -.13044191 | -.00411882 | -1.6657043 |
| Secondary Education | .00900482 | .12892804 | .00116097 | 1.8173017 | .02514205 | .12892804 | .00324151 | 1.3109106 |
| Higher | .0087813 | .39811928 | .00349601 | 5.4723897 | .00775496 | .39811928 | .0030874 | 1.2485847 |
| Subtotal | | | 0.00461612 | 7.22573986 | | | 0.00221009 | 0.893791 |
| **Working Status** | | | | | | | | |
| Had no Work (RC) | | | | | | | | |
| Had Work | .00352467 | 5.5172578 | -.06522455 | -.06928569 | .00451913 | 1.8275942 | .00451913 | 1.8275942 |
| **Wealth status** | | | | | | | | |
| Poorest (RC) | | | | | | | | |

*(Continued)*

**Table 4.** (Continued)

| | | | Hypertension | | | | Diabetes | |
|---|---|---|---|---|---|---|---|---|
| | | | Contribution to overall CIX = 0.06388445 (p<0.001) | | | | Contribution to overall CIX = 0.24727198 (p<0.001) | |
| | Elasticity | CIX | Absolute contribution | Percentage contribution | Elasticity | CIX | Absolute contribution | Percentage contribution |
| Poorer | .00408401 | -.41556635 | -.00169718 | -2.6566357 | -.00230154 | -.41556635 | .00095644 | .38679727 |
| Middle | .00984568 | -.01167131 | -.00011491 | -.17987477 | .01308682 | -.01167131 | -.00015274 | -.00015274 |
| Richer | .01978422 | .39371194 | .00778928 | 12.192768 | .0330336 | .39371194 | .01300572 | 5.2596837 |
| Richest | .02731484 | .7964429 | .02175471 | 34.05322 | .06229847 | .7964429 | .04961718 | 20.065831 |
| **Subtotal** | | | **0.0277319** | **43.40947753** | | | **0.0634266** | **25.71215923** |
| **Explained CI** | | | | **138.58018379** | | | | **38.14417404** |
| **Residual CI** | | | | **-38. 58018379** | | | | **-62. 14417404** |

BMI Stands for Body Mass Index, RC is to Reference Category, CIX means Concentration Index.

may result in an increase in sedentary lifestyle or a decrease in physical activity levels. It may also result in the adoption of Western lifestyles, resulting in a nutritional shift toward unhealthy food choices, such as increased intake of "fast foods" high in sugar and fat, both of which are known risk factors for overweight/obesity [58]. While overweight and obesity are risk factors for hypertension and diabetes [4, 5, 31, 59, 60]. However, due to increased access to health information and resources, there may be reduced burdens associated with behavioral and metabolic risk factors for NCDs among advantaged individuals [15]. Socioeconomic inequalities among NCD risk factors may be a result of previous interventions' inability to employ equity-based techniques for their reduction, as various populations may have varying levels of engagement in health programs or behavior change communication [4, 5, 16, 17, 59].

The chief findings of the study are that household wealth status accounted for approximately 25.71 and 43.41% of total inequality in hypertension and diabetes, respectively, in Bangladesh. While BMI contributed the substantial contribution to the inequality formation is 4.95 and 83.38%, respectively for hypertension and diabetes. In addition, urban areas contributed 4.56% inequality to increase diabetes among Bangladeshi inhabitants while administrative region contributed 4.76% of the inequality of hypertension. Similar findings have been reported in a variety of situations, demonstrating that those in the highest socioeconomic backgrounds are more likely to have diabetes and hypertension [18, 30, 61]. Due to a lack of access to health care, inadequate diabetes and blood pressure screening tools, lack of education, societal stigma, and inadequate health systems, a sizable section of the population remains undiagnosed [38, 40].

Obesity and overweight contributed 83.38% percent and 4.95%, to the total inequality in diabetes and hypertension, respectively. Along with contributing significantly to socioeconomic inequalities in the prevalence of diabetes and hypertension [18, 38, 62]. Of late, a rising trend toward becoming overweight or obese has recently been documented among Bangladesh's urban and affluent residents [62–64]. Obesity and overweight rates are rising because of causes such as the decline in physical activity owing to technologically intense job, watching television, using social media and the internet, and others. Another factor contributing to overweight and obese is the fact that those from affluent household in low-income nations are less likely to engage in physical activity, having much leisure times, which in turn results in greater weight gain and disease risk [35, 65]. Because physical activity frequently boosts oxygen consumption throughout the body and maintains blood glucose levels to protect the central

nervous system, it may help minimize the risk of diabetes and other noncommunicable diseases [18, 44].

Additionally, this study found that geographical variations in diabetes contributed considerably to diabetes inequality. Although the causes for this are unknown, certain places are predicted to have a disproportionately high rate of undiagnosed diabetes [40]. Though there were no significant socioeconomic differences had found for all divisions. Rangpur division contributed 3.73% inequality. This could be because of socioeconomic inequalities such as limited resources, income inequality, low level of education and social safety net programs, poor connectivity with the urban centres, distance of health facilities, fragile communication system and insufficiency or absence public facilities [4, 13, 66]. As a result, adopting administrative region-specific policies for addressing hypertension and diabetes in such areas should be addressed [18]. Further research on the risk factors that contribute to these geographical inequalities in Bangladesh is necessary.

According to the findings, achieving the target of halving premature NCD mortality in Bangladesh by 2030 may be attainable by leveraging development expenditures. Understanding the pattern of socioeconomic inequalities in risk factors for NCDs could aid in improving NCD preventive, development, and poverty reduction efforts, which are critical components of the global action plan for NCD prevention and control [15, 67]. The clear evidence that NCD risk factors influence disadvantaged individuals of LMICs may prompt development agencies dealing with these populations to alter their programs to address NCDs. The study's findings highlight the critical need for disaggregated data, which was stressed during the United Nations High Level Meeting on Noncommunicable Diseases [39]. The stage of economic growth, cultural factors, social, and health policies may all influence the socio-economic gradient of inequalities in NCD risk factors. People who live in distant or difficult-to-reach communities, as well as those with poor socioeconomic position, may have less access to health care for early diagnosis and treatment of NCDs than those who live in urban areas or have a higher socioeconomic status. Additional research on the emergence of inequality in chronic NCDs through time, as well as on the sociodemographic factors that drive inequality, is necessary to have a better understanding of the underlying causes and reasons for the current distribution of chronic NCDs in Bangladesh. The social and environmental factors that contribute to Bangladesh's inequitable distribution of disease have not been stated explicitly, and must be further investigated in view of the obvious prevalence of disparities between urban and rural areas.

The study's strengths and limitations are acknowledged. The study's strength is its generalizability to Bangladesh, as the analysis included nationally representative data from all divisions. Along with appropriate statistical methods for estimating the age-standardized prevalence of hypertension and diabetes in the sample, decomposition of the inequality measure and Lorenz estimates were performed using standard approaches. However, our study has some certain shortcomings. Due to the study's cross-sectional methodology, no causal association between risk factors could be established. Additionally, other risk factors (such as dietary habits, physical activity, family history of non-communicable diseases, and alcohol and smoking usage) were omitted from the data collection process might influence the study results. Therefore, further detailed cohort study is warranted.

## Conclusions

The present study concludes that diabetes and hypertension are more widespread in Bangladesh's urban areas among the well-off individuals. It is recognized that inequalities in wealth distribution and geographic location contribute to the burdens associated with NCDs risk

factors. This obvious distinction highlights the importance of creating focused intervention strategies to address the growing problem of NCDs and associated risk factors in these groups.

## Author Contributions

**Conceptualization:** Md. Ashfikur Rahman.

**Data curation:** Md. Ashfikur Rahman.

**Methodology:** Md. Ashfikur Rahman.

**Software:** Md. Ashfikur Rahman.

**Validation:** Md. Ashfikur Rahman.

**Visualization:** Md. Ashfikur Rahman.

**Writing – original draft:** Md. Ashfikur Rahman.

**Writing – review & editing:** Md. Ashfikur Rahman.

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
