## [Decision Letter · Decision Letter 0]

4 Jul 2022

PONE-D-22-09701Socioeconomic Inequalities in the Risk Factors of Noncommunicable diseases (Hypertension and Diabetes) Among Bangladeshi Population: Evidence based on population level data analysisPLOS ONE

Dear Dr. Rahman,

Thank you for submitting your manuscript to PLOS ONE. After careful consideration, we feel that it has merit but does not fully meet PLOS ONE’s publication criteria as it currently stands. Therefore, we invite you to submit a revised version of the manuscript that addresses the points raised during the review process.

We look forward to receiving your revised manuscript.

Kind regards,

Jennifer Annette Campbell

Academic Editor

PLOS ONE

Journal Requirements:

5. We note you have included a table to which you do not refer in the text of your manuscript. Please ensure that you refer to Table 2 in your text; if accepted, production will need this reference to link the reader to the Table.

Reviewers' comments:

Reviewer's Responses to Questions

**Comments to the Author**

1. Is the manuscript technically sound, and do the data support the conclusions?

Reviewer #1: Yes

Reviewer #2: Yes

Reviewer #3: Yes

Reviewer #4: Yes

2. Has the statistical analysis been performed appropriately and rigorously? 

Reviewer #1: Yes

Reviewer #2: Yes

Reviewer #3: Yes

Reviewer #4: Yes

3. Have the authors made all data underlying the findings in their manuscript fully available?

Reviewer #1: Yes

Reviewer #2: Yes

Reviewer #3: Yes

Reviewer #4: Yes

4. Is the manuscript presented in an intelligible fashion and written in standard English?

Reviewer #1: Yes

Reviewer #2: No

Reviewer #3: No

Reviewer #4: Yes

5. Review Comments to the Author

Reviewer #1: 1. Did the author exclude hypertension cases with pregnancy? Because hypertension cases may resolve after childbirth.

2. Author may look at individual association between risk factors for hypertension and diabetes.

3. Some studies have showed higher hypertension rate in working people since most of current jobs are desk jobs. Author found in this study that people with jobs have decreased risk of hypertension and diabetes. This is an interesting area to explore, e.g. job nature, working hour. If not explored, dissimilarities like this can be included in the discussion section.

4. The discussion section of the article is not well organized. The primary focus of the research question is 'socioeconomic inequality'. Therefore, while writing the discussion section, author may organize it by socioeconomic factors.

5. Author may link the study finding with literature more rigorously.

6. The website provided in the data availability section (https://dhsprogram.com/) do not share data with third party without permission. Hence, it is advisable to make the dataset for diabetes and hypertension visible / available to the editor for credibility of the analysis.

Reviewer #2: Dear Author

I must appreciate you for writing a nice manuscript focusing on NCDs including Diabetes and Hypertension.

However, I have some suggestions regarding your manuscript from the point of view of a reviewer.

Abstract: Nice, simple, and highly expressible.

Background: Excellent

Methodology: Nicely discussed and statistically sound.

Results: The tables and figures were well interpreted. Some important variables (added salt, working environment, food habit, family history of NCDs, lifestyle, smoking, alcohol, tobacco, etc. ) regarding NCDs are missing here, however, these limitations are manageable during working with the secondary dataset. I would like to add here that there is huge room to improve in the linguistic section, especially in sentence making, parts of speech, and run-on sentences.

Discussion: The article was nicely discussed and compared with the other related published articles. I think that linguistic improvement is also necessary for this section. I shall hereby suggest taking help from the professional body.

Conclusion: Briefly, the entire article is reflected in this section.

I must be thanking you once again for preparing this manuscript and have good luck with your manuscript.

Best Wishes

Reviewer.

Reviewer #3: 1. The paper is not well written. Some sentences are long with limited references. The justification of the study is poorly written there is necessity and room for improvement.

2. Methods section:

1. Data source is described to be the DHS. A little more specific details of how the data were obtained, what procedures were followed and long with what, if, a letter was required and for which datasets/variables data was accessed and how long it took would give readers more picture of the DHS data access.

3.Results:

Informative a bit lengthy. Also could use language and editorial revisions. some of the sentences were too long and difficult to understand the intent.

4.Conclusion:

Nothing new is found requires major revision, appears too small. Suggest some recommendations.

Reviewer #4: Dear authors,

Thank you for the opportunity to review your manuscript. I found the paper to be generally well written, and the topic to be interesting. My comments/concerns with the current manuscript are noted below.

Main Comments:

Methods:

1. Study population and survey design – what were the inclusion criteria used in order to arrive at the 12,290 sample size? Also, please clarify whether this represents the weighted or unweighted sample size.

2. Outcome variables – what was the reason for merging the patients with prediabetes and those with diabetes? Please clarify this within the text.

3. (Major) Independent variables (page 4, first sentence) – the first sentence is missing a citation. Also, more details on the variables included are needed as they are of considerable importance to the interpretation of later findings.

a. I encourage you to consider adding a conceptual model section that clearly outlines the categories of variables that influence your outcome variables, and then to list/explain the variables you include in order to adjust for each of these effects/channels within the context of your study.

b. Page 5: why are you including age covariates among your independent variables? I ask this because the abstract and statistical analysis section notes that your outcomes are already age-standardized.

c. Are the “wealth status” categories based on pre-set cutoffs, or upon wealth deciles that you create form your data? Please clarify this within the text.

4. Statistical Analysis:

a. How was age standardization of the data done? Please consider adding details to an appendix.

b. Page 7: the reference numberings do not align with those within the reference section at the end, please correct this.

c. Page 7: the term elasticity is mentioned, but not defined. Please add definition and explanation of this measure.

Results:

5. Table 1: it is said that values within parathesis are “(unweighted)” counts, however, it seems that they are percentages. Please check these. If they are indeed supposed to represent unweighted counts then why are they in decimal form and not presented as integers?

6. Prevalence of diabetes and hypertension section -- Please ensure sentences are complete without the information within parathesis, and also check punctuation placement (see e.g. end of second sentence).

7. On page 10 it is stated that “with age, the rate of hypertension increased …”, but was the hypertension rate not already age-standardized? Please clarify if this is referring to the unstandardized hypertension rate.

8. Page 12: please clarify what tests the p-values correspond to. I.e., please clarify the null and alternative hypotheses for these values.

9. Page 15: here it is stated that: “The column 'Elasticity' indicates the amount of change in the dependent variable (socioeconomic inequality in hypertension and diabetes) that occurs when the explanatory factors change by one unit.”. Please define the term elasticity. If this is referring to the traditional use within economics, then this is a unit-less measure, and as such, the end of this sentence should be revised accordingly.

10. Results/Discussion: in presenting the main findings as percentages, it is important to: (a) provide additional details within the text on how these are computed within Table 4; and (b) clarify to readers that these do not necessarily sum to 100%. While this might be clear to practitioners of these methods, given the broad readership of PLOS One, I think it is important to explain this within the text to avoid confusion.

Minor Comments:

Background:

11. Page 2, paragraph 1: the last sentence is unclear and needs to be revised.

Methods:

12. Statistical Analysis – the sentence starting with “Then, we assessed hypertension’s age… “ is incomplete. Please revise.

Results:

13. All tables – please add footnotes with explanation of all abbreviations under the table.

Discussion/Conclusions:

14. Discussion – page 19, paragraph 2, first sentence: please clarify what is meant by “…significantly lower among the less socioeconomic conditions…”.

6. PLOS authors have the option to publish the peer review history of their article (what does this mean?). If published, this will include your full peer review and any attached files.

Reviewer #1: **Yes: **Ishrat Binte Aftab

Reviewer #2: **Yes: **Goutam Kumar Acherjya, Assistant Professor (Medicine), Jashore Medical College, Jashore, Bangladesh.

Reviewer #3: No

Reviewer #4: No

---

## [Author Response · Author response to Decision Letter 0]

12 Jul 2022

all comments have been addressed which are available in the attachment named as Rebuttal.

---

## [Editor Report · Decision Letter 1]

9 Aug 2022

PONE-D-22-09701R1Socioeconomic Inequalities in the Risk Factors of Noncommunicable diseases (Hypertension and Diabetes) Among Bangladeshi Population: Evidence based on population level data analysisPLOS ONE

Dear Dr. Rahman,

Thank you for submitting your manuscript to PLOS ONE. After careful consideration, we feel that it has merit but still does not fully meet PLOS ONE’s publication criteria as it currently stands. Therefore, we invite you to submit a revised version of the manuscript that addresses the points raised during the review process.

Outstanding Edits Recommended Prior to Consideration: Reviewer 4's comments have not been adequately addressed. Specifically, we would like to see the following elements more carefully included in the manuscript: (1) additional details on certain data / measures / definitions are needed throughout; (2) the final (main) decomposition analyses depend on the variables included within the model, and as such, it seems particularly important for the authors to explain/motivate their choice of variables beyond what is currently done (it would additionally be beneficial for them to base these variable choices on a clear conceptual model/framework, somethings that is currently missing); (3) the interpretations of results are presented as percentages off of the overall CIX value, which needs to be clearly explained in order to avoid confusing many readers who might, for example, assume these sum to 100% (which they do not).Please note that while reviewers commented on the manuscript being well written, the grammatical errors that currently exist throughout this manuscript have to be addressed before it can be considered for publication. In addition, please remove the first person tense. We invite you to visit the PLOS Writing Center for free resources found here: https://plos.org/resources/writing-center/?utm_medium=website&utm_source=journalsite&utm_campaign=writingcenter&utm_content=submissionguidelines Additional specific edits noted by the editor: 1) Patient centered language is needed. For example, Line 102 “80% of the diabetic people reside” is not appropriate. See https://doi.org/10.2337/dci17-0041 The Use of Language in Diabetes Care and Education. 2) Line 106, please provide the date of the recent nationwide survey with some explanation as to why there would be such a significant rise in prevelance. For example, does the methodology vary and therefore did not adequately capture prevalence? 3) Line 162, please use phrasing consistent with the dataset description for diabetes oral medications and site reference. 4)Line 174, grammar. Had no working…this description appears to be incomplete. ==============================

We look forward to receiving your revised manuscript.

Kind regards,

Jennifer Annette Campbell, PhD, MPH 

Academic Editor

PLOS ONE

Journal Requirements:

PLOS ONE does not copyedit accepted manuscripts, so the language in submitted articles must be clear, correct, and unambiguous. We may reject papers that do not meet these standards.

If the language of a paper is difficult to understand or includes many errors, we may recommend that authors seek independent editorial help before submitting a revision. These services can be found on the web using search terms like “scientific editing service” or “manuscript editing service.”
---

## [Author Response · Author response to Decision Letter 1]

12 Aug 2022

all comments have been addressed adequately and provided in rebuttal section.

---

## [Editor Report · Decision Letter 2]

31 Aug 2022

PONE-D-22-09701R2Socioeconomic Inequalities in the Risk Factors of Noncommunicable Diseases (Hypertension and Diabetes) among Bangladeshi Population: Evidence Based on Population Level Data AnalysisPLOS ONE

Dear Dr. Rahman,

Thank you for submitting your manuscript to PLOS ONE. We appreciate your response to each concern and would like to consider for publication, however prior to moving forward there remain a few outstanding edits that are needed and are considered very minor. Therefore, we invite you to submit a revised version of the manuscript that addresses the points raised.

We look forward to receiving your revised manuscript.

Kind regards,

Jennifer Annette Campbell, PhD, MPH

Academic Editor

PLOS ONE

Journal Requirements:

Additional Editor Comments:

Introduction:

Line 93

Please rephrase to “…80% of people living with diabetes reside in…”

Line 99

Please rephrase to “Due to demographic transitions and….”

Methods:

Line 122 - BDHS spell out initially 

Line 155 – Change “considered diabetes” to “….were classified as having diabetes” 

Statistical Analysis

Line 189-191

Please change this statement to read:  “As a result, the prevalence of hypertension and diabetes were normalized in the same standard population in order to eliminate or reduce the effect of participant age and sex distribution discrepancies.”

Line 193-194

Please change this statement to this statement: “Following that, a log-binomial regression model with survey weights and explanatory variables with p-values (0.05) was used…”

Line 224

Please add “the”: Among the population…

Results

Line 280 

Please rephrase to: “A total of 12,290 were included in the final analysis.”

Line 288

Please do not use the word objects to refer to human subjects. Please rephrase to “…among the sample population” 

Line 290 refers to “women” the preceding sections refer to “females”. For consistency please refer to all as “women” throughout the manuscript. 

Line 307 please correct, current sentence states “log binomial passion” 

Line 365 refers to diabetes and prediabetes. The manuscript as a whole is focused on diabetes and hypertension. Please address if prediabetes is also being examined. 

Line 435 please remove “to” from this sentence so that it reads, “may be responsible…”

Line 437 please add “are” so that this sentence reads “…hypertension and diabetes are significantly lower…”

Line 443-444 please address grammar by rephrasing to: “While overweight and obesity are risk factors for hypertension and diabetes…”

Line 471-472 please address grammar by rephrasing to: “Another factor contributing to overweight and obese…”

Line 492, pattern is used twice. Please remove duplicate. 

Line 510, please address, is this referring to plural studies or one study. If plural, please state “Previous studies have tried...” If singular, please state, “A previous study tried…” please add references to site which prior studies have tried. 

Line 528, please address grammar by stating: “The present study concludes….”

---

## [Author Response · Author response to Decision Letter 2]

1 Sep 2022

Revision has been provided in the rebuttal section

---

## [Editor Report · Decision Letter 3]

8 Sep 2022

Socioeconomic Inequalities in the Risk Factors of Noncommunicable Diseases (Hypertension and Diabetes) among Bangladeshi Population: Evidence Based on Population Level Data Analysis

PONE-D-22-09701R3

Dear Dr. Rahman,

We’re pleased to inform you that your manuscript has been judged scientifically suitable for publication and will be formally accepted for publication once it meets all outstanding technical requirements.

Kind regards,

Jennifer Annette Campbell, PhD, MPH

Academic Editor

PLOS ONE
---

## [Editor Report · Acceptance letter]

12 Sep 2022

PONE-D-22-09701R3 

Socioeconomic Inequalities in the Risk Factors of Noncommunicable Diseases (Hypertension and Diabetes) among Bangladeshi Population: Evidence Based on Population Level Data Analysis 

Dear Dr. Rahman:

I'm pleased to inform you that your manuscript has been deemed suitable for publication in PLOS ONE. Congratulations! Your manuscript is now with our production department. 

Kind regards, 

on behalf of

Dr. Jennifer Annette Campbell 

Academic Editor

PLOS ONE